# EZH2 Inhibition as New Epigenetic Treatment Option for Pancreatic Neuroendocrine Neoplasms (PanNENs)

**DOI:** 10.3390/cancers13195014

**Published:** 2021-10-07

**Authors:** Simon Leonhard April-Monn, Valentina Andreasi, Marco Schiavo Lena, Martin Carl Sadowski, Corina Kim-Fuchs, Michelle Claudine Buri, Avanee Ketkar, Renaud Maire, Annunziata Di Domenico, Jörg Schrader, Francesca Muffatti, Claudio Doglioni, Stefano Partelli, Massimo Falconi, Aurel Perren, Ilaria Marinoni

**Affiliations:** 1Institute of Pathology, University of Bern, 3008 Bern, Switzerland; simon.april@pathology.unibe.ch (S.L.A.-M.); andreasi.valentina@hsr.it (V.A.); martin.sadowski@pathology.unibe.ch (M.C.S.); michelle.buri94@gmail.com (M.C.B.); avanee3894@gmail.com (A.K.); renaud.maire@pathology.unibe.ch (R.M.); annunziata.didomenico@gmail.com (A.D.D.); 2Graduate School for Cellular and Biomedical Sciences, University of Bern, 3012 Bern, Switzerland; 3Pancreatic Surgery Unit, Pancreas Translational and Clinical Research Center, San Raffaele Scientific Institute, 20132 Milan, Italy; muffatti.francesca@hsr.it (F.M.); partelli.stefano@hsr.it (S.P.); falconi.massimo@hsr.it (M.F.); 4Faculty of Medicine and Surgery, Vita-Salute San Raffaele University, 20132 Milan, Italy; doglioni.claudio@hsr.it; 5Unit of Pathology, San Raffaele Scientific Institute, 20132 Milan, Italy; schiavolena.marco@hsr.it; 6Department of Visceral Surgery and Medicine, University Hospital Bern, University of Bern, 3008 Bern, Switzerland; corina.kim-fuchs@insel.ch; 7Department of Medicine, University Medical Center Hamburg-Eppendorf, 20251 Hamburg, Germany; jschrade@uke.de; 8Bern Center for Precision Medicine, University & University Hospital of Bern, 3008 Bern, Switzerland

**Keywords:** pancreatic neuroendocrine neoplasms, EZH2 (Enhancer of Zest homolog), tumor treatment, epigenetic treatment, histone modification

## Abstract

**Simple Summary:**

Pancreatic neuroendocrine neoplasms (PanNENs) represent 3% of pancreatic neoplasms. Available therapies can induce stable disease only for a minority of patients. Overall survival ranges from 10 years for well-differentiated neuroendocrine tumors to as little as 10 months for more aggressive carcinomas (NECs). It has been shown that epigenetic aberrations are relevant for the development and progression of PanNENs. We found that increased expression of the methyl transferase EZH2 correlated with higher tumor grade and advanced disease status. Inhibition of EZH2 in vitro reduced cell viability and proliferation of PanNEN cell lines as well as of patient-derived islet-like tumoroids. Similarly, inhibition of EZH2 in a PanNEN transgenic mouse model reduced tumor burden. Our data indicate that EZH2 inhibition should be further investigated/considered as an epigenetic treatment for patients with high-grade PanNENs.

**Abstract:**

Pancreatic neuroendocrine neoplasms are epigenetically driven tumors, but therapies against underlying epigenetic drivers are currently not available in the clinical practice. We aimed to investigate EZH2 (Enhancer of Zest homolog) expression in PanNEN and the impact of EZH2 inhibition in three different PanNEN preclinical models. EZH2 expression in PanNEN patient samples (*n* = 172) was assessed by immunohistochemistry and correlated with clinico-pathological data. Viability of PanNEN cell lines treated with EZH2 inhibitor (GSK126) was determined in vitro. Lentiviral transduction of shRNA targeting EZH2 was performed in QGP1 cells, and cell proliferation was measured. Rip1TAG2 mice underwent GSK126 treatment for three weeks starting from week 10 of age. Primary cells isolated from PanNEN patients (*n* = 6) were cultivated in 3D as islet-like tumoroids and monitored for 10 consecutive days upon GSK126 treatment. Viability was measured continuously for the whole duration of the treatment. We found that high EZH2 expression correlated with higher tumor grade (*p* < 0.001), presence of distant metastases (*p* < 0.001), and shorter disease-free survival (*p* < 0.001) in PanNEN patients. Inhibition of EZH2 in vitro in PanNEN cell lines and in patient-derived islet-like tumoroids reduced cell viability and impaired cell proliferation, while inhibition of EZH2 in vivo in Rip1TAG2 mice reduced tumor burden. Our results show that EZH2 is highly expressed in high-grade PanNENs, and during disease progression it may contribute to aberrations in the epigenetic cellular landscape. Targeting EZH2 may represent a valuable epigenetic treatment option for patients with PanNEN.

## 1. Introduction

Pancreatic neuroendocrine neoplasms (PanNENs) represent 3% of pancreatic tumors. PanNENs are a heterogeneous group of neoplasms with varying clinical behaviour, ranging from indolent, low-grade pancreatic neuroendocrine tumors (PanNETs) to malignant, highly aggressive neuroendocrine carcinomas (NECs). The WHO 2019 classification separates PanNETs from PanNECs based on cellular differentiation, genetic patterns, and histo-morphological features. The grading system, based on the percentage of Ki67-positive, proliferating tumor cells, further separates PanNETs into G1, G2, and G3 [1]. While G1 PanNETs may have an overall survival (OS) of more than 10 years, OS for G2 PanNET is roughly 6 years [2]. On the other hand, high-grade PanNENs show worse survival outcomes, with patients diagnosed with NECs surviving less than 10 months [3].

Well-differentiated G1 and G2 PanNETs present mutations in *MEN1*, *DAXX*, and *ATRX* in almost 40% of patients, while 15% carry mutations in genes encoding members of the mTOR pathway [4,5]. PanNECs are frequently mutated in *KRAS*, *SMAD4*, and *TP53*, and they additionally often display a loss of Rb1 [6]. Clinical management of PanNETs and PanNECs is challenging. Medical treatment schedules for advanced and progressing PanNETs commonly include somatostatin analogues (SSAs) as first-line therapy and either Everolimus, Sunitinib, Temozolomide, Streptozocin, or peptide receptor radionuclide therapy (PRRT) as second-line treatments. (Pan)NEC patients typically receive platinum-based chemotherapy as first-line therapy [7]. Unfortunately, none of these therapies is able to induce stable disease in a predictable way. Therefore, better and more personalized treatments are urgently needed.

Recently, the importance of epigenetics for the development and progression of PanNETs has become evident [8]. DAXX, ATRX, and MENIN are all involved in chromatin structure remodelling and maintenance [9]. Additionally, loss of H3K36me3 and ARID1A (AT-Rich Interaction Domain 1A), a member of the SWI/SNF family, has been described in T3/T4 and metastatic PanNETs [10]. Chromatin structure organization is dictated by specific histone modification patterns, which in turn are tightly regulated by specific enzymes. Histone modifications are fundamental in maintaining cell identity and in regulating processes such as cellular differentiation. Alteration of histone modification patterns and their regulating enzymes have been widely described in different cancer types. Hence, targeting such modifications has become an attractive treatment option. 

EZH2 (Enhancer of Zest homolog) is a histone-lysine N-methyltransferase enzyme and a member of the polycomb-group proteins. As catalytic subunit of the polycomb repressive complex (PRC2), it is responsible for the trimethylation (me3) of lysine 27 (K27) on histone 3 (H3) to promote gene silencing [11,12]. Notably, EZH2 is found highly expressed in stem cells and downregulated in adult tissues (reviewed in [11]). EZH2 and the PRC2 complex regulate the expression of several genes involved in cell differentiation. There are many downstream pathways possibly contributing to cell transformation dependent on EZH2 alteration. Indeed, EZH2 downstream targets include *CDKN2A*, E-cadherin, *FOXC1*, as well as DNA repair pathways [11]. Overexpression of EZH2 has been described in several cancer types and has been associated with poor prognosis and aggressive disease [13]. Given the evidence for EZH2 as a cancer driver, the development of EZH2-specific inhibitors has been an active area of investigation. Several EZH2 inhibitors have shown promising results in vitro, and several clinical trials have been successfully conducted [14,15,16]. Here, we show that high EZH2 expression is associated with advanced status and high aggressiveness of disease in PanNENs. Inhibition of EZH2 in PanNEN cell lines and patient-derived islet-like tumoroids impaired cell proliferation in vitro. Similarly, treatment of Rip1TAG2 mice, a transgenic PanNEC mouse model, with EZH2 inhibitor reduced tumor burden. 

Altogether, our findings suggest that EZH2 inhibition may represent a potentially promising treatment option, especially for high-grade PanNENs.

## 2. Materials and Methods

### 2.1. Patient Collective

Patient characteristics are shown in Table 1. 

The study was approved by the Swiss cantonal authorities (Kantonale Ethikkomission Bern, Ref.-Nr. KEK-BE 105/2015) and the Italian ethics commission (Comitato Etico, CE 252/2019). All patient materials were used according to the human research act and had signed an institutional form of broad consent. Immunohistochemistry was performed on PanNET next-generation Tissue Micro Arrays (ngTMAs), including for 129 patients that underwent surgery at the Inselspital, Bern, Switzerland, between 1990 and 2020 (reported in part in [17]) and 43 additional patients who underwent surgery at S. Raffaele Hospital, Milan, Italy, between 2017 and 2020. All cases were reclassified according to WHO 2017 criteria [18]. TNM staging was based on the eight edition of the UICC/AJCC [19]. 

In brief, 2.5 µm sections from ngTMAs or whole blocks were used for immunohistochemistry of EZH2 (1:50, Cell Signaling, 5246) and H3K27me3 (Dilution, Cell Signaling, C36B11). The immunostainings for all antigens were performed with an automated staining system (Leica Bond RX; Leica Biosystems, Nunningen, Switzerland). Antigen retrieval was performed by heating Tris30 buffer at 95 °C for 30 min. The primary antibodies were incubated for 30 min at the specified dilutions. Visualization was performed using a Bond Polymer Refine Detection kit, using DAB as chromogen (3,3’-Diaminobenzidine). EZH2 scoring was performed using QuPath software (open source software for digital pathology image analysis) by automatically counting the number of tumor cells expressing EZH2 [20]. The mean nuclear optical density was used to define positive and negative tumor cells. H3K27me3 staining was scored as negative, heterogeneous, and positive. For both EZH2 and H3K27me3 scorings, only nuclear staining was considered positive. DAXX and ATRX immunohistochemistry were performed as previously described [17]. 

### 2.2. Cell Culture

The BON1 cell line was provided by E.J. Speel, Maastricht, Netherlands, in 2011. The QGP1 cell line was purchased from the Japanese Health Sciences Foundation, Osaka, Japan, in 2011. The NT3 cell line was a kind gift from J. Schrader and cultured as described [21]. Short tandem repeat (STR) analysis by PCR was performed for all cell lines (QGP1 in 2011, 2016, and 2020; BON1 in 2014, 2016, and 2020; NT3 in 2018 and 2020). QGP1 cells were authenticated. A BON1 and NT3 profile does not exist yet, but the profile of these cells did not match any known profile of cancer cell lines, thus excluding contamination from other lines. In addition, expression of the specific neuroendocrine markers chromogranin A and synaptophysin was routinely tested by IHC. For NT3, the cell culture flasks were coated with collagen IV for better attachment of the cells. BON1 cells were cultured in DMEM/F12 medium (Sigma), whereas QGP1 and NT3 cells were cultured in RPMI 1640 medium (Sigma). For all cell lines, the medium was supplemented with 10% FBS, 100 IU/mL penicillin, and 0.1 mg/mL streptomycin, and cells were kept in a humidified incubator at 5% CO2 and 37 °C. Additionally, growth factors EGF (Gibco PHG0314) and FGF2 (Gibco PHG0024) were added to NT3 growth medium. After thawing, cells were cultured for approximately two months. 

### 2.3. In Vitro Drug Treatment 

#### 2.3.1. MTT Assay

For treatment with GSK126 (Selleckchem), cells were plated in 96 wells and treated with 0.62 µM, 2.5 µM, 6.255 µM, and 12.5 µM, 25 µM, 25 µM, and 100 µM of GSK126 diluted in DMSO. Control cells were treated with 0.5% DMSO. The cells were incubated with 100 μL 10% MTT solution at 37 °C in 5% CO2 for 40 min. After MTT removal, 200 µL of DMSO and 25 μL of Sorensen solution were added to lyse the cells. The intensity of the color was measured as absorbance at 570 nm on a Microplate Reader (SpectraMax, Molecular Devices, San Jose, CA, USA).

#### 2.3.2. IncuCyte Real-Time Cell Confluence

Real-time cell proliferation as a function of cell confluence was measured by live microscopy with an IncuCyte S3 system (Essen BioScience, Newark, NJ, USA). BON1 and QGP1 cells were seeded in their respective cell culture medium at 5000 cells/well in 96-well Essen ImageLockTM plates (Essen BioScience, Newark, NJ, USA). After 48 h of culture, cells were treated in technical replicates (*n* = 3) with vehicle control (DMSO) or indicated concentrations of GSK126; plates were transferred to the IncuCyte S3 system, and images were acquired every 2 h for 4 days with a 10× objective. Measurements were normalized to the mean confluence (~25%) of all wells at t = 0. Representative images for t = 48 h are shown in Appendix A (see also the Appendix A).

### 2.4. Western Blotting

Non-histone proteins were extracted using RIPA buffer, and protein concentrations were measured using the Bradford assay. Histones were extracted using an acid extraction protocol. After washing with PBS, cells were scraped off in 30 μL 0.4 M HCl and incubated on ice for 30 min with intermittent vortexing. The lysates were centrifuged at 10,000 rpm for 10 min at 4 °C. Supernatant was collected. To this, 360 μL of ice-cold acetone was added and the tubes were kept at −20 °C overnight. The day after, lysates were centrifuged at 10,000 rpm for 10 min at 4 °C. The acetone containing supernatant was discarded and the histone-enriched pellet was resuspended in 30 μL 4 M urea + Pi buffer. Protein concentration was measured using the Bradford assay. Histones were loaded onto precast gradient gels (4–15%) from Biorad (#4568085). Non-histone proteins were loaded onto gels (12%) made as per the manufacturer’s instructions by mixing stacker and resolver solutions from Biorad (#1610180). After running, gels were activated in a Biorad Chemidoc MP system. Transfer was done on to PVDF membranes using a Trans Blot Turbo system from Biorad at 1.3A, 25 V, for 7 min. Post-transfer, total proteins were imaged with a Biorad Chemidoc MP system. After 1 h blocking, incubation with primary antibodies was performed overnight at 4 °C, followed by washing steps and incubation with secondary antibodies (DyLight 650 conjugate goat anti-rabbit and DyLight 550 conjugate goat anti-mouse (ImmunoReagents) and peroxidase-conjugated AffiniPure donkey anti-rabbit and donkey anti-mouse (Jackson ImmunoResearch)) for 1 h at room temperature. Chemiluminescent or fluorescent signals were detected using a ChemiDoc MP System (Biorad). Total protein expression for quantification of specific protein expression was measured by use of the stain-free gel technology and imaged with the Chemidoc MP System [22]. The primary antibodies EZH2 (1:1000, Cell Signaling, 5246), H3K27me3 (1:2000, Cell Signaling 9733), H3 total (1:5000 Abcam ab12079), and GAPDH (1:5000, Millipore MAB 374) were diluted in 5% BSA-TBST. Band intensity was measured using ImageJ and the area size calculation tool of the plotted lane (square pixel).

### 2.5. EZH2 Silencing 

Short hairpin RNA (shRNA) against EZH2 (TRCN0000040074, TRCN0000040075), as well as a nontargeting shRNA control (SHC002), were delivered with a lentivirus expressing vector pLKO.1 (all from Sigma, MISSION shRNA). Lentivirus production and transduction were performed as described previously [23]. Cells were selected with 1.5 μg/mL puromycin for 3–4 days. Knockdown efficiency was validated by immunoblotting of respective proteins.

### 2.6. In Vivo Experiments

Rip1TAG2 (C57BL/6) mice were kindly provided by G. Christofori (Basel, Switzerland). All experimental protocols were reviewed and approved by the Cantonal Veterinary Office of Bern (Bern, Switzerland). Mice were fed with food enriched in glucose starting from 10 weeks of age. Vehicle control (20% Captisol in sterile H20) and GSK126 (100 mg/kg, ST061, Selleckchem) was administrated daily by i.p. injection for three weeks. GSK126 stock was dissolved in 20% modified cyclodextrin (Captisol^®^, LGND, USA) and sterile H2O. In brief, Captisol was acidified to pH 4 using 1N acetic acid before adding GSK126 stock solution. The drug solution was stirred for two hours at 4 °C using sterile magnets. The solution was then sonicated for 1 min at 40% amplitude at 37 °C in an ultrasonic water bath, ensuring temperature did not exceed 40 °C. The final drug solution was adjusted to pH 4.5 using 1N acetic acid. After i.p. application (200 ul per 20 g body weight), animal health status was monitored daily. At 13 weeks of age, animals were sacrificed and dissected. Tumor numbers (>1 mm) were counted by visual inspection. The tissues were then fixed in formalin overnight and embedded in paraffin. FFPE tissue was used for tumor burden quantification/assessment using QuPath software [20]. Digital-scanned consecutive IHC tissue sections were first pre-processed in the built-in visual stain editor using default settings for estimation of stain vectors. Total tumor area and all areas containing endocrine (islet) cells were manually annotated and verified by a board-certified pathologist (SL. M.) on the first H&E tissue slide. These annotations were transferred onto (all) consecutive tissue slides for consistency. A watershed cell segmentation followed by positive cell detection was performed using customized/optimized parameters and individual thresholds for each specific IHC staining. Detection results were extracted from QuPath and imported into R environment for data analysis.

### 2.7. Primary Cells Treatment

For primary cell isolation, viability measurement, micro-cell block manufacture, and quantification, we followed the described protocol [24]. Fresh human PanNET tissue was obtained from patients diagnosed with PanNETs undergoing surgery at the Inselspital Bern, Switzerland, or at the Pancreatic Surgery Unit, Pancreas Translational and Clinical Research Center, San Raffaele Scientific Institute, Milan, Italy, as previously described [24]. Cryopreserved tumor tissues of six PanNET patients were used for in vitro drug screening. Patient characteristics are summarized in Table 2.

#### 2.7.1. Primary Cell Culture 

Isolated primary PanNET cells were maintained in AdvDMEM + GF medium (DMEM-F12, 5% FBS, Hepes 10 mM, 1% L-glutamine (200 mM), 1% penicillin (100 IU/mL), 1% streptomycin (0.1 mg/mL), 1% amphotericin B (0.25 mg/mL) (Merck, Switzerland), 20 ng/mL EGF, 10 ng/mL bFGF (Thermo Fisher Scientific, USA), 100 ng/mL PlGF, 769 ng/mL IGF-1 (Selleckchem, USA)) and in 24-well Corning^®^ Costar^®^ ultra-low attachment (ULA) plates (Corning, USA) (500 µL/well, 3–5 × 10^5^ cells/well) in a humidified cell incubator (21% O2, 5% CO2, 37 °C). For drug screening, cells were resuspended in fresh AdvDMEM + GF medium supplemented with 123 µg/mL growth-factor-reduced Matrigel^®^ (Corning, USA) and plated in 96-well ULA plates (50 µL/well, 3–4 × 10^3^ cells/well). 

#### 2.7.2. Primary Cell Isolation and Culture

Cells were isolated and cultured as previously reported [24]. In brief, washed pieces of 1 mm^3^ were dissociated in digestion medium (10 mg/mL collagenase IV (Worthington, USA), 0.25% Trypsin-EDTA (Sigma-Aldrich, Switzerland), and 0.2 mg/mL DNase (Roche, Switzerland) in advanced DMEM-F12, Hepes 10 mM, 1% L-glutamine, 1% penicillin-streptomycin-amphotericin B) using a gentle MACSTM dissociator (Miltenyi Biotec, Switzerland). Cells were filtered through a 70 µm strainer to remove collagen debris, and red blood cells were lysed with ACK lysis buffer (Thermo Fisher, Scientific, USA). After mechanical fibroblast removal and single-cell dissociation, cells were resuspended and maintained in AdvDMEM + GF medium. After 2 days of recovery phase, cells were counted and resuspended in fresh AdvDMEM + GF medium supplemented with growth-factor-reduced Matrigel and plated in 96-well ULA plates (3–4 × 10^3^ cells/well). 

#### 2.7.3. Viability Measurement in Islet-Like Tumoroids

The RealTime-Glo™ MT Cell Viability (RTG) assay (Promega, Switzerland) was used to continually monitor cell viability of 3D human primary PanNET cultures. The RTG assay was performed according to the manufacturer’s instructions, and luminescence was measured in an Infinite^®^ 200 PRO plate reader (Tecan, Switzerland).

#### 2.7.4. Micro-Cell-Blocks (MCBs) from Islet-Like Tumoroids

Islet-like tumoroids corresponding to 3–5 × 10^4^ cells were collected (either directly on the day of isolation (D0) or from the 96-well ULA plate at the end of drug screening (D12)). Cells were captured in plasma-thrombin clots and fixed, counterstained with Hematoxylin, and embedded in paraffin. The, 2.5-µm-thick serial sections were stained as described above. Scans were acquired with a Panoramic 250 (3DHistech, Hungary) automated slide scanner at 20× magnification. Images were acquired using QuPath software [20].

#### 2.7.5. Curve Fitting and Drug Sensitivity Data

Drug-response curve data consisted of six to nine DMSO-positive controls, six no-cell-negative controls, and five drug-response points. A 5-point, 625-fold concentration range was used to calculate reliable absolute IC50 values [25]. For IC50 calculation, RLU values from a 7 day treatment were weighted and normalized using 6 h RTG-baseline measurements for each well, as described earlier [24]. Data points were fitted in a four-parameter linear (4PL) regression model with two constraints, Top = 100% and Bottom = 0%, to estimate the corresponding IC50 [26,27]. Visualization was performed in R environment. 

### 2.8. Correlation and Survival Analysis

Statistical analyses were performed with GraphPad Software. Unpaired or paired t-tests were used to compare groups. When the normality assumption was not met, the Mann–Whitney test or Kruskal–Wallis test were used to compare continuous variables between groups. Contingency tables were analyzed using Fisher’s exact test. Cut-offs to define low, intermediate, and high EZH2 expression were defined using the median and the third quartile of EZH2 distribution as a continuous variable. Survival probability was estimated according to the Kaplan–Meier method. The log-rank test was used to compare disease-free survival between EZH2 categories. Sample size (n) refers to biological replicates unless otherwise stated. *, *p* < 0.05; **, *p* < 0.01; ***, *p* <0.001; ****, *p* < 0.0001.

## 3. Results

### 3.1. EZH2 Expression in PanNEN Correlates with Advanced Disease Status and Features of Aggressiveness 

To evaluate the expression of EZH2 in PanNEN tissues we performed immunohistochemistry (IHC) on a tumor microtissue collective of 172 patients who underwent surgery for PanNENs (Table 1).

As shown in Figure 1A, EZH2 expression in PanNENs was highly heterogeneous. The percentage of EZH2-positive tumor cells was scored for each patient. Based on EZH2 positivity, the samples were then divided in three categories, using the median (1.5%) and the third quartile (3%) of EZH2 distribution as cut-offs: <1.5% of positive tumor cells (EZH2^low^); 1.5% ≤ x ≤ 3% of positive tumor cells (EZH2^intermediate^); and >3% of positive tumor cells (EZH2^high^). In 79% of PanNENs (*n* = 136/172), the percentage of tumor cells positive for EZH2 was ≤3%. Only 21% of tumors (*n* = 36/172) showed a percentage of EZH2-positive tumor cells >3%. No significant differences in terms of EZH2 expression were observed according to the time of surgery (*p* = 0.590). In agreement with EZH2 function in regulating genes involved in cell cycle, EZH2 expression correlated with the Ki67 proliferative index (*p* < 0.001). Median Ki67 progressively increased across EZH2 categories, ranging from 1.5% (IQR 1; 4%) in samples with EZH2^low^ to 3.5% (IQR 1.5; 7%) in samples with EZH2^intermediate^ and up to 15% (IQR 5; 40%) in samples with EZH2^high^ (Appendix A). When functioning tumors (*n* = 32/172) were excluded, the correlation between EZH2 expression and Ki67 proliferative index remained statistically significant (EZH2^low^: median Ki67 2% (IQR1; 4%), EZH2^intermediate^: median Ki67 5% (IQR 2; 9.5%), EZH2^high^: median Ki67 15% (IQR 7.5; 45%), *p* < 0.001). In line with this—as reported in other tumor types—increased EZH2 expression was significantly associated with a higher tumor grade (*p* < 0.001). G1 PanNETs were the most represented group within the EZH2^low^ category (*n* = 55/80, 69%), whereas G2 PanNETs were prevalent when EZH2 expression was intermediate (*n* = 32/56, 57%) or high (*n* = 21/36, 58%), as depicted in Figure 1B. A significant association between EZH2 expression and tumor grade was confirmed after exclusion of patients with functioning neoplasms (*p* < 0.001). Overall, 14 out of 15 G3 PanNENs showed positivity for EZH2 in >3% of tumor cells (Figure 1B). Indeed, we observed that PanNECs had >60% EZH2-positive cells in the majority of cases. Additionally, using publicly available RNA-sequencing data, we confirmed in silico that *EZH2* gene expression is higher in G2 and G3 tumors compared to G1 [5] (Figure 1C). 

Significantly higher protein expression of EZH2 was also observed for patients with T3–T4 tumor stage compared to those with T1–T2 (*p* = 0.004) as well as in presence of nodal (*p* = 0.008) and distant metastases (*p* < 0.001) (Table 3 and Figure 1D).

Patients with EZH2^high^ showed distant metastases in 51% (*n* = 18/35) of cases compared to 12% (*n* = 10/80) of patients with EZH2^low^. Interestingly, higher EZH2 positivity was found in samples negative for DAXX/ATRX (*p* = 0.014) (Table 3). 

Follow-up data were available for 105 patients (*n* = 98 PanNETs, *n* = 7 PanNECs) and the median follow-up was 37 months (IQR 18–60 months). The recurrence rate in the whole study cohort was 30% (*n* = 32/105). Patients with EZH2^high^ showed also a shorter disease-free survival compared to those with EZH2^low^ and EZH2^intermediate^ (*p* < 0.001) (Figure 1E). Patients with EZH2^low^ and EZH2^intermediate^ showed better DFS compared to those with EZH2^high^ (*p* = 0.016), and also after excluding patients with functioning neoplasms (*n* = 32). This statistically significant difference in survival was also observed when patients with NECs were included in the analysis, as shown in (Appendix A). No significant differences in H3K27me3 levels were observed between the different categories (Appendix A).

### 3.2. Inhibition of EZH2 in PanNEN Reduced Cell Viability and H3k27me3 Levels 

Given the expression of EZH2 in PanNEN and especially its higher expression in PanNEC patient samples, we investigated if pharmacological inhibition of EZH2 would impair cell growth and induce cell death in vitro. To this purpose, we first measured EZH2 protein expression in three PanNEN cell lines, BON1, QGP1 (both with mutations indicative of PanNECs), and NT3 (from a high-grade G2 PanNET), by Western blotting (Figure 2A). As expected from their origins, BON1 and QGP1 expressed high levels of EZH2, while it was expressed at lower levels in NT3 cells. Next, we pharmacologically targeted EZH2 with the competitive inhibitor GSK126. Monitoring of proliferation as a function of cell confluence in real-time revealed that GSK126 inhibited growth of QGP1 and BON1 cells in a dose-dependent manner (Figure 2B), with cell clusters being visibly smaller and containing fewer cells after 48 h of treatment (Appendix A). After longer treatment periods and at higher GSK126 doses (25 µM and 50 µM), cells showed morphological signs of apoptotic cell death (loss of cell–cell contacts, membrane blebbing, cell shrinkage; data not shown and Appendix A). Since loss of epigenetic activity might require longer treatment periods to establish a cellular phenotype, we measured cell viability using MTT assays after 3 and 6 days. As shown in Figure 2C, all three cell lines showed a decrease in cell viability in a dose- and time-dependent manner. All three cell lines displayed similar sensitivities to different drug concentrations with similar IC50 values: 18.0 μM (BON1), 23.1 μM (QGP1), and 15.4 μM (NT3) for 3 days and 8.0 μM (BON1), 15.8 μM (QGP1), and 5.8 μM (NT3) for 6 days of treatment, respectively.

Thus, our data demonstrate that EZH2 inhibition with GSK126 is cytotoxic in PanNEN cells in vitro. In order to confirm that GSK126-mediated cytotoxicity was associated with loss of EZH2 methyltransferase activity, we quantified the tri-methylation levels of EZH2′s histone downstream target, H3K27 (H3K27me3), after GSK126 treatment by Western blotting. We confirmed that H3K27me3 levels of QGP1 and BON1 cells were significantly and equally decreased by all tested concentrations of GSK126 in BON1 and QGP1 after 6 days of GSK126 treatment (Figure 2D,E). Due to a low number of NT3 cells and insufficient protein quantity after six days of GSK126 treatment, H3K27me3 levels in NT3 were assessed after three-day treatment only. However, this showed a significant reduction of H3K27me3 levels in a dose-dependent manner (Figure 2D,E). Together, these data demonstrate that EZH2 inhibition by GSK126 reduced its methyltransferase activity and affected cell viability in PanNEN cells in vitro.

### 3.3. Silencing of EZH2 in High-Grade PanNEN Cell Lines Impaired Cell Growth 

To rule out any off-target effects from pharmacological GSK126 treatment, we silenced EZH2 by lentiviral transduction in the high-grade PanNEN cell line QGP1. Cells were transduced using lentivirus vectors of two different shRNA (40074 and 40075) and one scrambled shRNA control. ShRNA 40074 was less efficient than the shRNA 40075 and induced an EZH2 knockdown of 43% at day one and 29% at day seven of selection, respectively, while shRNA 40075 induced a knockdown of 70% at day one, which was reduced to 54% at day seven (Figure 3A). The downregulation was confirmed by IHC on cell blocks as well (data not shown). To investigate the role of EZH2 depletion on cell growth, we produced a growth curve using the MTT assay for 4 days after selection. Cells transduced with Sh-40075 showed an almost complete stop of proliferation, while cells transduced with Sh-40074 grew at a reduced rate when compared to scrambled controls (Figure 3B). Notably, the inhibition of proliferation was proportional to the efficiency of the knockdown. Altogether, these results strongly support a critical role for EZH2 in promoting cell survival and proliferation in high-grade PanNEN cell lines. 

### 3.4. Anti-EZH2 Treatment of Rip1TAG2 Mice Reduced H3K27me3 Levels and Tumor Burden

Following up on this, we assessed the therapeutic effect of EZH2 inhibition in vivo in the Rip1TAG2 mouse model [28]. In this model, the simian virus 40 (SV40) large T-antigen (Tag) oncogene is expressed under the control of the rat insulin gene promoter (Rip), leading to multifocal development of insulin-producing β-cell carcinomas (insulinoma) in the islets of Langerhans in the pancreas [28]. Effects of EZH2 inhibition in vivo were assessed by comparing GSK126-treated mice (*n* = 6, 3F/3M) with littermate control mice (*n* = 6, 3F/3M) over the time course of three weeks starting from 10 weeks of age (Figure 3C). Consecutive formalin-fixed paraffin-embedded (FFPE) sections from resected pancreas were analyzed by a pathologist (SL. M.), and islets were annotated as normal islets (Ns), proliferative islets (PIs), hyperplastic islets (HPs), and tumors (Ts) (adapted from [29]) (Appendix A). IHCs were quantified digitally using QuPath software. In this model, we observed an increase in EZH2 expression along different stages of tumorigenesis (Figure 3D). Inhibition of EZH2 decreased trimethylation of H3K27 in proliferative and hyperplastic islets as well as tumors, confirming the on-target effect of GSK126 (Appendix A). We detected an unexpected but slight decrease in EZH2 expression in hyperplastic islets and tumors in treated mice, but the expression levels remained high in abnormal islets of both treated and untreated mice (Figure 3D and Appendix A). We observed a significant reduction in tumor burden in GSK126-treated mice (*p* = 0.00039) (Figure 3E) and a tendency towards a reduction of the number of tumors (Appendix A). No differences in the Ki67 percentage of positive cells and cleaved caspase 3 were detected (Appendix A).

### 3.5. Treatment of Patient-Derived PanNET Tumoroids with EZH2 Inhibitors Reduced Cell Viability

Although EZH2 is highly expressed in PanNECs, we found that a subset of G2 PanNETs also express EZH2, albeit at a lower level. To assess if EZH2 inhibition may be a therapeutic option for PanNET patients, we treated patient-derived islet-like tumoroids isolated from six PanNET patients (two liver metastases and four primary tumors) with EZH2 inhibitor (GSK126) using our previously reported screening pipeline [24]. Patient characteristics are summarized in Table 2. Islet-like tumoroids were treated with GSK126 in a five-point, 625-fold concentration treatment scheme (0.06 µM, 1.60 µM, 0.32 µM, 8.00 µM, and 40 µM). Tumoroids from different patients showed distinctive drug sensitivities (Figure 4A and Table 2).

Micro-cell-blocks of two representative islet-like tumoroids *before* and *after* treatment are shown in Figure 4B. To correlate the response measured in vitro with EZH2 expression of the corresponding tissue of origin, we performed IHC. EZH2 staining was scored as described above. EZH2 expression was low in three samples (<1.5% of positive tumor cells), intermediate in one sample (1.5% ≤ x ≤ 3% positive tumor cells), and high (>3% of positive tumor cells) in the two remaining cases (Appendix A). No clear correlation was observed between EZH2 expression in the tumor tissue and drug sensitivity; however, the sample number was relatively small.

## 4. Discussion

In this work we demonstrated that a subset of PanNENs expressed EZH2 and that its expression highly correlated with higher tumor grade and disease stage. We showed that inhibition of EZH2 in vitro and in vivo in PanNEC and PanNET models reduced growth, cell survival, and tumor burden. Altogether, our results suggest that EZH2 inhibition may be a novel epigenetic treatment option for PanNEN patients.

PanNET development seems to be mainly driven by epigenetic changes; several lines of evidence demonstrated a possible progressive accumulation of epigenetic aberrations along PanNET expansion [8,9,10]. Epigenetic changes involve histone and DNA modifications, which can result in profound phenotypic changes. These epigenetic events are inherently reversible; hence, targeting such modifications in cancer has become a promising option. A plethora of drugs targeting specific enzymes responsible for histone modifications, such as methylation, acetylation, or phosphorylation, are either already in the clinics or in clinical trials, and many others are in preclinical development [30,31,32]. Targeting EZH2 is among one of the most promising epigenetic therapies in cancer treatment but has not yet been evaluated in PanNEN [14,15,16].

We found that EZH2 is particularly highly expressed in PanNECs and G3 PanNETs, with lower expression being present in G2 and G1 PanNETs. Based on these observations, we explored the option of treating both PanNECs and PanNETs with EZH2 inhibitor GSK126, using different in vitro and in vivo models.

EZH2 is expressed in many cancer types in correlation with advanced disease stage and high proliferation index [33]. Indeed, EZH2 expression is regulated by the pPB-E2F pathway, and it has been shown to be critical for cell replication. Hence, EZH2 is universally recognized as a marker of proliferation and a bona fide oncogene [34]. This is in agreement with our observation that EZH2 expression is highly correlated with Ki67 positivity in PanNENs. In a small study including 30 patients, increased EZH2 expression was described in human PanNETs with synchronous metastases compared to those with metachronous ones. However, no correlation with tumor grade was reported [35].

The EZH2 locus was found amplified in a subset of insulinomas, and overexpression of EZH2 was reported to induce replication of human beta cells as well as other normal islet cells [36]. In mouse models, EZH2 epigenetically represses CDKN2A/p16INK4A in pancreatic beta cells, and it is required for beta cell proliferation in juvenile mice [37].

Our results suggest a crucial role for EZH2 in mediating PanNEN cell proliferation. Silencing of EZH2 in PanNET cells by EZH2 inhibitor or siEZH2 showed a strong reduction in cell proliferation. This cytostatic effect most likely occurred via cell-cycle arrest, since it has been shown previously that gene silencing of EZH2 in cancer cell lines stopped proliferation and increased the number of cells in G1 and G2 [34]. EZH2 inhibition in PanNEN cell lines and in Rip1TAG2 mice resulted in reduction of global H3K27me3 levels, likely releasing the H3K27me3 gene repression at certain loci. Since GSK126 is highly selective, EZH2 methyltransferase-inhibition (see Selleckchem REF#S7061) off-target effects via other human methyltransferases are unlikely. However, due to EZH2′s diverse molecular functions—from our data—we cannot delineate the exact mode of action. In PanNEN cell lines we noticed a reduction of H3K27me3 levels already at GSK126 dosages that showed no obvious impact on cell viability or proliferation, suggesting that other EZH2 effector functions might be relevant as well. Indeed, besides H3K27me3, the PRC2 complex methylates non-histone protein substrates as well. In addition, EZH2 via a PRC2-independent function methylates or directly interacts with other proteins, activating downstream pathways [33]. Via these three different mechanisms, EZH2 works as a hub for several pathways that are crucial for cancer development, such as cell-cycle progression, autophagy, apoptosis, DNA repair cell development, and lineage differentiation [33]. The lack of correlation between H3K27me3 and EZH2 expression in human tissue suggests that EZH2 may indeed function independently from PRC2 in PanNENs.

EZH2 expression in PanNENs increased with tumor grade and the majority of PanNECs showed positivity in more than 60% of tumor cells. Given the high percentage of EZH2-positive cells in PanNECs, EZH2 inhibition may represent a promising therapeutic strategy for these tumors for which no targeted treatment is currently available.

In support of this, we found that EZH2 inhibition in Rip1TAG2 mice reduced tumor burden. Rip1TAG2 mice present with tumors that share similarities with human PanNECs in terms of morphology and aggressiveness [28]. Due to the transgenic large T-antigen, both P53 and RB are inactivated, similarly to PanNECs, which often present with *TP53* mutation and RB loss [28]. While we could see a reduction in tumor burden, we did not find clear changes in Ki67 and caspase-3, leaving open some questions on how EZH2 inhibition impairs tumor progression. While we observed reduction in tumor burden, we also observed a trend towards a reduction in the number of tumors. EZH2 expression in RipTag2 tumors increased with tumor size and animal age, suggesting that EZH2 inhibition may affect growth of late-stage tumors with higher EZH2 expression rather than of early-stage and small tumors. Similarly, EZH2 inhibition has been shown to reduce tumor burden and tumor growth in several preclinical models, such as lung cancer and lymphoma mouse models [38,39]. Interestingly, we found that low-grade PanNETs also express EZH2, albeit at lower levels. We recently established a protocol for cultivating patient-derived PanNET cells from fresh- and cryopreserved tumor tissue, which allows drug screening ex vivo [24]. Using this model, we tested the EZH2 inhibitor GSK126 on PanNET patient samples. Interestingly, we observed differences in GSK126 sensitivity among patients, suggesting a specific patient effect, despite lower EZH2 levels in lower grade PanNETs. These results suggest that EZH2 inhibition can also be relevant for the treatment of a subset of low-grade PanNETs, possibly in combination with other therapies.

Increasing evidence has recently demonstrated that EZH2 inhibition in combination with other treatments potentiates the antitumor effect of standard therapies. For example, EZH2 inhibition enhanced the effect of Temozolomide (TMZ) in TMZ-resistant glioblastoma cell lines [40].

Overall, our results indicate that EZH2 inhibition shows anti-tumoral effects in in vitro, in vivo, and ex vivo PanNEN models. EZH2 inhibition may represent a novel epigenetic treatment option for high-grade PanNEN.

## 5. Conclusions

In conclusion, we demonstrated that high EZH2 expression in PanNENs correlated with high grade, tumor stage, presence of metastases, and shorter disease-free survival and that EZH2 inhibition impaired cell viability and tumor burden. Notably, EZH2 expression was extremely high in highly proliferating PanNECs. Our data indicate that EZH2 inhibition may represent a novel, promising treatment option, especially for high-grade PanNENs.

## Figures and Tables

**Figure 1 cancers-13-05014-f001:**
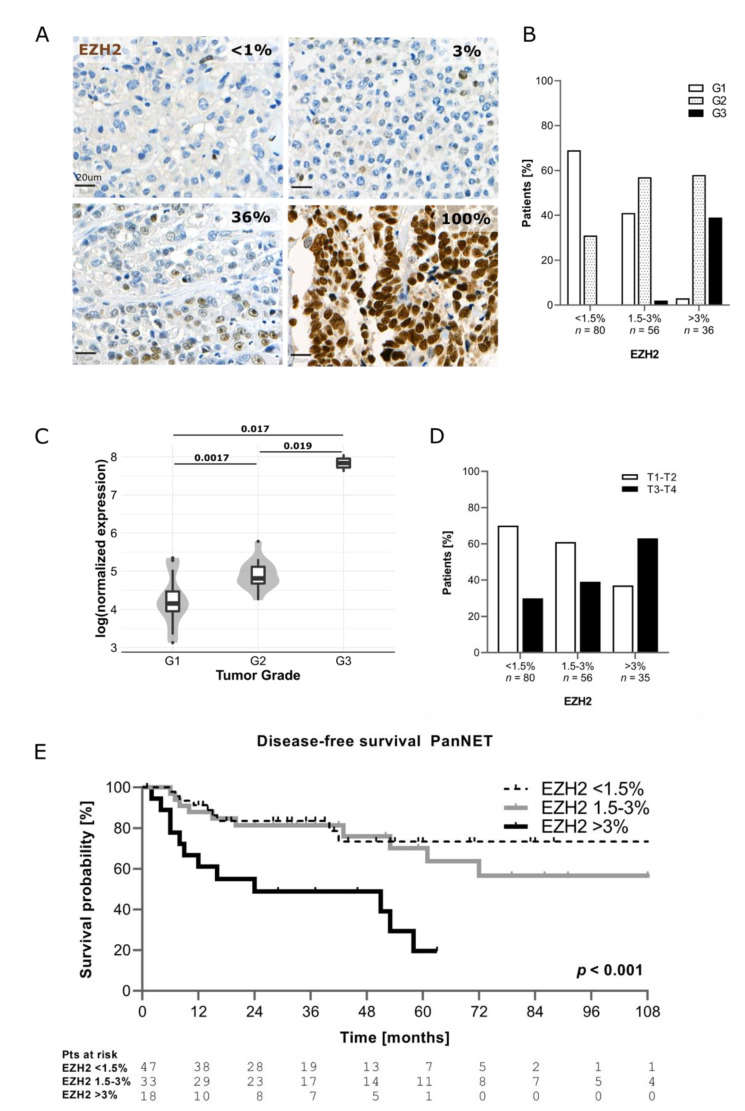
(**A**) Example of EZH2 expression in human tissue. (**B**) Correlation between EZH2 expression and tumor grade. (**C**) Correlation between EZH2 mRNA level and grade (data from Scarpa et al. 2017). (**D**) Correlation between EZH2 and T stage (T stage missing (*n* = 1)). (**E**) Comparison of disease-free survival between patients with low, intermediate, and high EZH2 expression (only PanNETs were included).

**Figure 2 cancers-13-05014-f002:**
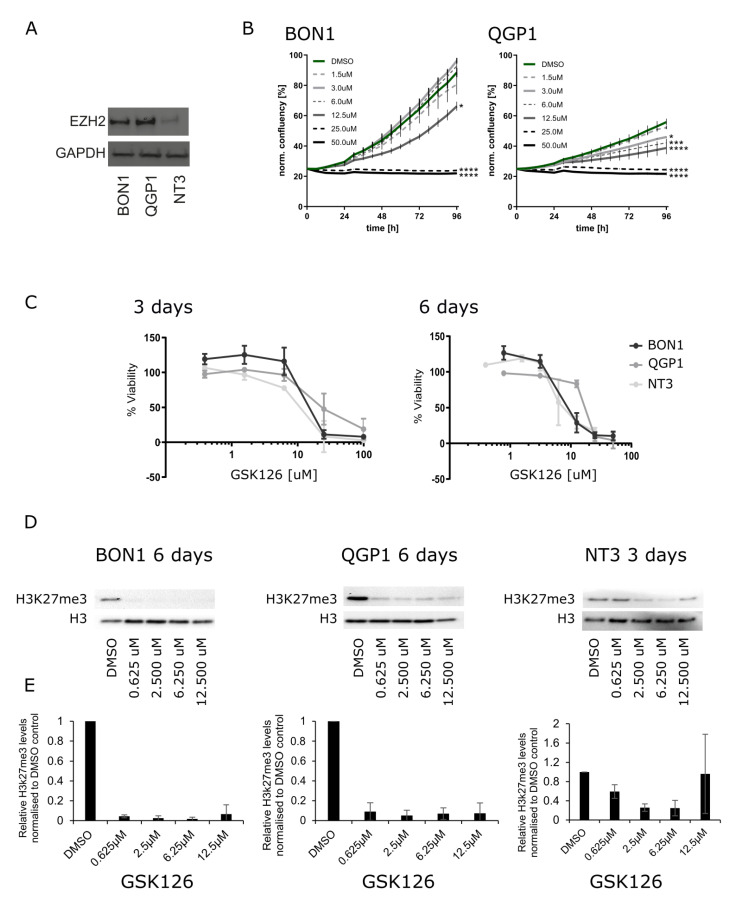
(**A**) Western blotting of EZH2 expression in PanNET cell lines: BON1, QGP1, and NT3. (**B**) IncuCyte S3 proliferation analysis of BON1 and QGP1 cells treated for 96 h with vehicle control (DMSO) and the indicated concentrations of GSK126. This is a representative result of two independent experiments. (**C**) MTT assays after EZH2 treatment with 0.62 µM, 2.5 µM, 6.255 µM, 12.5 µM, 25 µM, and 100 µM of GSK126 after 3 and 6 days. (**D**) Representative Western blotting of H3k27me3 after 6 days of treatment with 0.62 µM, 2.5 µM, 6.255 µM, and 12.5 µM GSK126 in BON1 and QGP1 and 3 days for NT3. (**E**) Quantification of H3K27me3 levels normalized to H3 total based on three replicates.

**Figure 3 cancers-13-05014-f003:**
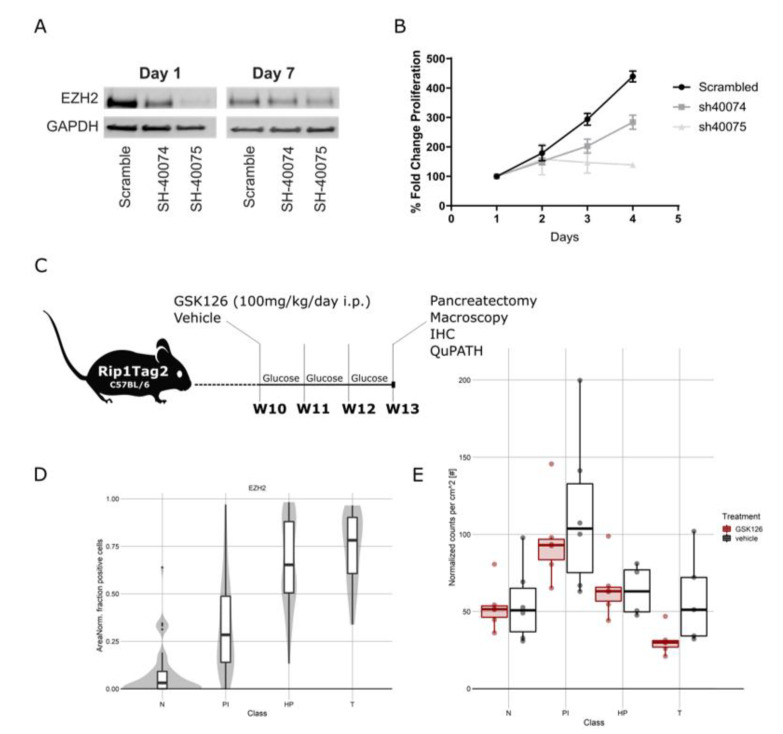
(**A**) Representative Western blot of EZH2 knockdown in QGP1 cells with SH40074, SH40075, and scrambled controls at 1 and 7 days after transduction. (**B**) Growth curve after transduction with SH40074, SH40075, and scrambled controls in QGP1. (**C**) Schematic representation of treatment of Rip1TAG2 mice. (**D**) EZH2 expression in Rip1TAG2 mice at different tumor stages: normal islet (N), proliferative islets (PI), hyperplastic islet (HP), and tumor (T). (**E**) Tumor burden in mice treated with EZH2 inhibitor and vehicle daily for 3 weeks from 10 weeks of age. Mice treated with EZH2 inhibitor presented a reduced tumor burden.

**Figure 4 cancers-13-05014-f004:**
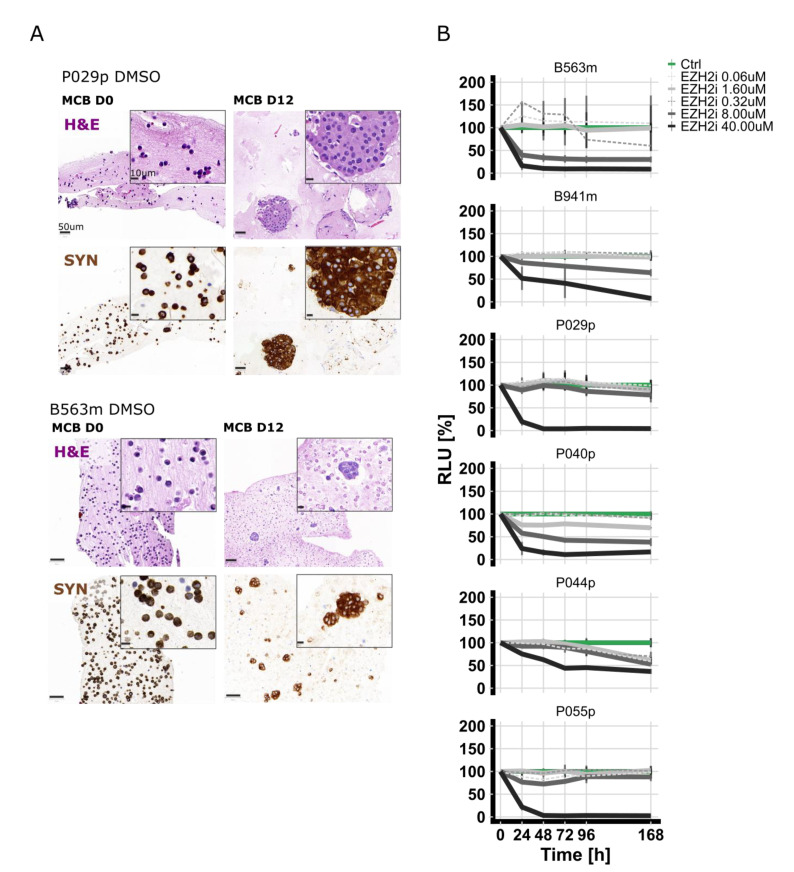
(**A**) In vitro viability curves using the metabolic surrogate assay RealTime-Glo (RTG) in 3D human primary PanNET culture treated with DMSO (control (Ctrl)) and GSK126 for 7 days. Data were first normalized per-well using a RTG baseline measurement for each individual well and then normalized to the average of the corresponding DMSO control of the respective day. Data represent means ± SEM (*n* = 1 per patient, three technical replicates). RLU, relative luminescence unit. (**B**) Micro-cell-block of two representative samples. IHC of synaptophysin and H&E staining of samples from the day of isolation and DMSO-treated samples 12 days post-isolation.

**Table 1 cancers-13-05014-t001:** Clinico-pathological features of patients submitted to surgery for pancreatic neuroendocrine neoplasms (PanNENs).

Variable	*n* = 172 (%)
Gender	
Male	92 (53)
Female	80 (47)
Age, years	59 (49; 69) *
Tumor function	
Non-functioning	140 (81)
Insulinoma	29 (17)
Gastrinoma	2 (1)
Glucagonoma	1 (1)
Tumor size, cm	3 (2.4; 4.1) *
T stage **	
T1–T2	103 (60)
T3–T4	68 (40)
N stage **	
N0	77 (46)
N1	67 (40)
Nx	24 (14)
M stage **	
M0	128 (75)
M1	43 (25)
Tumor grade	
NET G1	79 (46)
NET G2	78 (45)
NET/NEC G3°	15 (9)
Ki67, %	3 (1.5; 8) *
DAXX/ATRX **	
Negative	59 (36)
Positive	107 (64)

* Expressed as median (interquartile range). ** T stage missing (*n* = 1), N stage missing (*n* = 4), M stage missing (*n* = 1), DAXX/ATRX status missing (*n* = 6). *n* = 5 NET G3, *n* = 10 NEC G3.

**Table 2 cancers-13-05014-t002:** Patient characteristics of treated islet-like tumoroids.

Patient	Gender	Age	Grade	Ki67	Stage	DAXX/ATRX	EZH2	Tumor Site	In Vitro Sensitivity
mP029	Female	55	NET G2	4%	II	Lost	6.3%	Primary	+
mP040	Female	55	NET G2	10%	II	Preserved	3%	Primary	+++
mP044	Female	18	NET G2	18%	III	Lost	1.3%	Primary	+
mP055	Female	69	NET G2	8%	III	Lost	0.3%	Primary	+
aP321	Male	66	NEC G3	50%	IV	Lost	23%	Liver metastasis	++
aP476	Male	65	NET G2	15%	IV	n.a.	0%	Liver metastasis	+++

NET, neuroendocrine tumor; NEC, neuroendocrine carcinoma; n.a., not available; + lower sensitivity; ++ intermediate sensitivity; +++ higher sensitivity.

**Table 3 cancers-13-05014-t003:** Correlation between EZH2 expression and clinico-pathological features.

Variables	EZH2 <1.5% *n* = 80	EZH2 1.5–3% *n* = 56	EZH2 >3% *n* = 36	*p*
T Stage				0.004
T1–T2	56 (70)	34 (61)	13 (37)	
T3–T4	24 (30)	22 (39)	22 (63)	
N stage				0.008
N0	39 (50)	28 (51)	10 (29)	
N1	24 (30)	21 (38)	22 (65)	
Nx	16 (20)	6 (11)	2 (6)	
M stage				<0.001
M0	70 (88)	41 (73)	17 (49)	
M1	10 (12)	15 (27)	18 (51)	
DAXX/ATRX				0.014
Preserved	57 (74)	34 (63)	16 (46)	
Lost	20 (26)	20 (37)	19 (54)	

T stage missing (*n* = 1), N stage missing (*n* = 4), M stage missing (*n* = 1), DAXX/ATRX status missing (*n* = 6).

## Data Availability

The data presented in this study are available in the article.

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
