# Peer review of "EZH2 Inhibition as New Epigenetic Treatment Option for Pancreatic Neuroendocrine Neoplasms (PanNENs)"

_cancers, 2021, doi:10.3390/cancers13195014_

Round 1
Reviewer 1 Report
This manuscript by Simon Leonhard et.al described the role of EZH2 in pancreatic neuroendocrine neoplasms (PanNENs) progression and demonstrated EZH2 inhibition as a therapeutical strategy for PanNENs treatment. In this paper, GSK126 inhibitor was used in vitro and in vivo to reduce tumor cell viability by decreasing methyltransferase activity of EZH2. This manuscript is coherent and quite well-written.
There are some issues should be further clarified as below:
Major:
- EZH2 is an epigenetic silencer to methylation of H3K27me3, meantime, is a gene activator independent of PRC2 complex. How to know the inhibitor GSK126 used here is specifically targeting its silencer function but not its activator function in some other pathways? In some other words, how about the specificity of GSK126?
- What is the mechanism behind GSK126? Just inhibit methyltransferase activity on H3K27me3? Is there any effect on histone deacetylase inhibition? It should be further illustrated.
Minor:
- In Fig 2B, the line color is very difficult to distinguish, please change it.
- In Fig 4B, please change the line color to make it clear.
Author Response
Please see the file attached

Reviewer 2 Report
This paper is a well-written and an interesting original article concerning the influence of the inhibition of EZH2 on PanNEN. A differentiation was made into the correlation between the expression and the tumor stage and grading, the effect of the blockade in vitro and, in addition, the therapeutic effect in a mouse model. Overall, there was an increased EZH2 expression at a higher tumor stage and at a higher G stage. Inhibition affected viability and proliferation.
Overall, the results of this study are of interest. However, there are some critical drawbacks to be resolved.
The examined clinical samples summarized a time from 1990 to 2020. Was there a lower level of EZH2 expression, especially with older samples, with possible falsification of the result?
Figure 1 E shows the disease-free survival. No information was given on the follow-up data in the text.
The PanNEN included both functional and non-functional tumors. In the case of entirely different tumor characteristics, including the often benign insulinomas, a differentiation or exclusion of the functionally active tumors can be discussed.
Author Response
Plase see the file attached
